# Enhancing Adhesion and Reducing Ohmic Contact through Nickel–Silicon Alloy Seed Layer in Electroplating Ni/Cu/Ag

**DOI:** 10.3390/ma17112610

**Published:** 2024-05-28

**Authors:** Zhao Wang, Haixia Liu, Daming Chen, Zigang Wang, Kuiyi Wu, Guanggui Cheng, Yu Ding, Zhuohan Zhang, Yifeng Chen, Jifan Gao, Jianning Ding

**Affiliations:** 1School of Materials Science and Engineering, Jiangsu University, Zhenjiang 212013, China; 2112005014@stmail.ujs.edu.cn (Z.W.); liuhx@ujs.edu.cn (H.L.); ggcheng@ujs.edu.cn (G.C.); dingyu_uj@163.com (Y.D.); 2State Key Lab of Photovoltaic Science and Technology, Trina Solar Co., Ltd., Changzhou 213031, China; daming.chen@trinasolar.com (D.C.); zigang.wang@trinasolar.com (Z.W.); kuiyi.wu@trinasolar.com (K.W.); yifeng.chen@trinasolar.com (Y.C.); jifan.gao@trinasolar.com (J.G.); 3Institute of Technology for Carbon Neutralization, Yangzhou University, Yangzhou 225127, China

**Keywords:** Ni/Si alloy, annealing temperature, adhesion, plating, n-TOPCon solar cell

## Abstract

Due to the lower cost compared to screen-printed silver contacts, the Ni/Cu/Ag contacts formed by plating have been continuously studied as a potential metallization technology for solar cells. To address the adhesion issue of backside grid lines in electroplated n-Tunnel Oxide Passivating Contacts (n-TOPCon) solar cells and reduce ohmic contact, we propose a novel approach of adding a Ni/Si alloy seed layer between the Ni and Si layers. The metal nickel layer is deposited on the backside of the solar cells using electron beam evaporation, and excess nickel is removed by H_2_SO_4_:H_2_O_2_ etchant under annealing conditions of 300–425 °C to form a seed layer. The adhesion strength increased by more than 0.5 N mm^−1^ and the contact resistance dropped by 0.5 mΩ cm^2^ in comparison to the traditional direct plating Ni/Cu/Ag method. This is because the resulting Ni/Si alloy has outstanding electrical conductivity, and the produced Ni/Si alloy has higher adhesion over direct contact between the nickel–silicon interface, as well as enhanced surface roughness. The results showed that at an annealing temperature of 375 °C, the main compound formed was NiSi, with a contact resistance of 1 mΩ cm^−2^ and a maximum gate line adhesion of 2.7 N mm^−1^. This method proposes a new technical solution for cost reduction and efficiency improvement of n-TOPCon solar cells.

## 1. Introduction

Benefiting from carbon peaking and carbon neutrality strategies, the development of new energy has become increasingly important. Crystalline silicon solar cells are highly favored due to their high conversion efficiency and good stability [1,2]. Among various types of crystalline silicon solar cells, N-TOPCon solar cells have become the most competitive solar cells [3]. However, this cell type faces the challenge of high silver paste consumption during the metallization process. Screen printing (SP) metallization is a critical step in the production process of crystalline silicon solar cells. However, it is accompanied by the issue of high silver consumption [4]. Reportedly, the highest proportion of costs related to the production of crystalline silicon solar cells (not including silicon wafer prices) are attributed to the use of expensive silver paste. The silver consumption of n-TOPCon solar cells is about 110–120 mg per piece. As a result, it is crucial to explore alternatives to screen-printing processes in order to effectively decrease the cost of metallization [5].

At present, the main metallization methods reported to replace screen printing silver paste are as follows: electroplating (electrochemical deposition metallization) [6,7,8], laser transfer printing [9,10], inkjet printing [11,12,13], etc. Electroplating has emerged as a main research subject among these methods, owing to its significant cost reduction in metallization by employing copper electrodes. This is because copper (Cu 580 S/m) offers similar electrical conductivity to silver (Ag 630 S/m), but at a fraction of the price [14,15]. However, it should be noted that copper has a high diffusion rate in silicon and acts as a deep-level impurity and high recombination center, making it susceptible to the formation of Cu_3_Si compounds that could adversely affect the carrier lifetime in silicon [16,17] The current solution is to electroplate an Ni intermediate layer between the copper layer and silicon substrate, preventing direct contact between the Cu and underlying Si, thereby avoiding the formation of Cu_3_Si [18].

Nevertheless, the Si/Ni/Cu gate lines have low adhesion since the Ni layer and Si substrate are merely physically connected [19,20,21]. Fortunately, by performing thermal silicidation at different temperatures, chemical reactions can occur between nickel (Ni) and silicon (Si), forming silicides such as the Ni_2_Si phase, NiSi phase, and NiSi_2_ phase [22,23,24,25,26]. Among these, the NiSi phase not only improves the adhesion of gate lines and reduces contact resistance but also enhances the thermal stability of cells. The findings presented by Kluska et al. [27] suggest that the thermal silicidation anneal caused by NiSi alloy improves gate line adhesion. Furthermore, research by Kale et al. suggests that the NiSi/Si interface exhibits better thermal stability compared to the Ni/Si interface [28]. In addition, direct annealing of Si-Ni-Cu can result in the formation of Si-NiSi-Ni-Cu. However, due to the significant difference in the diffusion rates of Cu, Ni, and Si at the same temperature, Kirkendall voids may form at the interfaces, which is detrimental to achieving high thermal stability in the solar cell [29]. In view of this, in this study, the NiSi alloy layer was prepared on the Si substrate using an annealing process, followed by electroplating Ni and Cu successively on its surface, ultimately producing Si-NiSi-Ni-Cu gate line contacts.

In this work, the Ni layer was deposited on the back of the crystalline silicon solar cell and then annealed in the temperature range of 300 °C to 425 °C to form a seed layer (NixSiy). Subsequently, Ni and Cu were sequentially electrodeposited on the seed layer, and, finally, the Si-NiSi-Ni-Cu contact structure was constructed. In order to further explore the phase information and its formation mechanism in the Ni/Si alloy layer, we used X-ray diffraction (XRD), Raman spectroscopy (Raman), X-ray photoelectron spectroscopy (XPS), and other test methods for detailed analysis. In order to evaluate the effect of the seed layer (NixSiy) on the performance of solar cells, we further adapted the Sun-Voc test method. The test results show that the Ni/Si alloy formed by annealing at 375 °C is closer to the NiSi phase, which not only has lower contact resistance but also exhibits an excellent grid-bonding force. This discovery is of great significance for improving the performance of solar cells.

## 2. Experiment

### 2.1. Fabrication of Solar Cells

All experimental wafers use n-type monocrystalline silicon wafers with an area of 158.75 mm × 158.75 mm and a thickness of 170 μm. Firstly, the random pyramid structure was prepared on the surface of the silicon wafer, and then the p-n junction was formed on the front side by the diffusion method. Then, ultra-thin SiO_2_ and n-type polysilicon (n-poly) were prepared by low-pressure chemical vapor deposition (LPCVD) to form a TOPCon contact structure. The AlOx film with a certain thickness was prepared on the front of the solar cell as the front passivation layer by atomic layer deposition (ALD). Then, the front and back SiNx films were deposited by plasma-enhanced chemical vapor deposition (PECVD), and the front AlOx and the front and back SiNx layers were opened using a UV-ps laser. The width of the front laser contact opening was about 7 μm, and the width of the back laser contact opening was about 40 μm.

The nickel layer was deposited on the back of n-TOPCon using electron beam deposition (0.5 A s^−1^ deposition rate). The samples were annealed at various temperatures of 300–500 °C in an inert nitrogen (N_2_) atmosphere. The unreacted Ni was removed with an etching solution (mixed solution of H_2_SO_4_ and H_2_O_2_). The back plating process sequence includes a single-sided low-concentration HF pretreatment to remove any oxide layer within the laser contact opening (LCO). Subsequently, nickel and copper plating was performed by light-induced plating (LIP). Subsequently, the solar cell was flipped using almost the same process sequence for depositing Ni and Cu. However, this time, the plating does not require light because the pn junction is in forward bias (forward bias (FBP)). The front also appears in the electroless silver bath. The thickness of the Ni layer is about 0.5 μm, the thickness of the Cu layer is about 7 μm, and the thickness of the Ag layer is 0.3 μm (where Cu is used as a conductive layer, Ag is used as an anti-oxidation layer, and it is convenient for subsequent welding to make components). The flow chart of the seed layer crystalline silicon solar cell is shown in Figure 1.

### 2.2. Characterization Methods

The surface morphology of the seed layer was observed by scanning electron microscopy, and the chemical composition of the sample was determined with an Energy-Dispersive Spectrometer (EDS) (Carl Zeiss AG, Oberkochen, German). The surface roughness of the seed layer was determined with a high-resolution optical profiler (Zeta^TM^-3.0) (Tokyo, Japan). The X-ray diffraction (XRD) measurements were performed using a Cu Kα radiation diffractometer with a scan rate of 0.01° s^−1^ in 2θ range of 5–80° to obtain detailed phase formation information. The phase transition for different types of Ni silicide was characterized through X-ray Photoelectron Spectroscopy (XPS, Thermo Fisher Scientific, Waltham, MA, USA) and Raman Spectroscopy (Thermo Fisher Scientific, Waltham, MA, USA). For measuring the contact resistance of the seed layer stack, the Transmission Line Model (TLM) technique was employed. For the adhesion measurement of the seed layer, a peel force test was conducted by using a universal testing machine. Meanwhile, the electrical properties of n-TOPCon solar cells with seed layers with different annealing conditions were investigated by evaluating from Suns-Voc (Sinton FCT650) (Boulder, CO, USA) measurements.

## 3. Result and Discussion

In order to evaluate the surface appearance of the seed layer, SEM tests were carried out. Figure 2a is the SEM diagram of the annealing temperature of 375 °C. It can be seen from the figure that the surface has an obviously uneven particle feeling. Appendix A shows the surface morphology of different annealing temperatures, all of which show rough graininess. The production of nickel–silicon alloy compounds or the ablation state brought on by the laser process could be the origin of this morphology, which would increase the surface roughness. We performed EDS (as shown in Figure 2b) to get more insight into the elemental makeup of the annealed samples. The findings of the research demonstrate that the elements Ni and Si are dispersed equally across the sample’s surface. The oxygen detected on the surface of the sample is caused by the oxidation of the sample surface in the air. To further confirm that the formed compound is a nickel–silicon alloy, we performed XRD, Raman, and XPS tests.

After annealing at different temperatures, Ni and Si form different phases, such as Ni_2_Si, NiSi, and NiSi_2_ [30,31]. Among the different nickel silicide phases formed by the seed layer, the NiSi phase is the lowest resistivity phase in the silicide phase. From the XPS diagram of the seed layer annealed at 400 °C shown in Figure 2c, it can be seen from the full peak that both the Ni peak and the Si peak exist, and the Ni2p and Si2p peaks are most suitable at 852.7 and 98.8 eV, respectively. The Ni2p peaks of Ni_2_Si, NiSi, and NiSi_2_ are all around 853 eV, and the positions of Si2p peaks are also very close [32,33]. Since the observed Si2p peak is close to the Si2p peaks of NiSi and Ni_2_Si at 98.85 eV and 98.75 eV [34], the XPS needs to be combined with XRD analysis. In addition, Si2p has a prominent peak at 103.3 eV, corresponding to the Si2p peak of SiO_2_, which may be caused by oxidation on the surface (Appendix A) [35].

Based on XRD measurements conducted under various annealing conditions, the atomic ratio of the compound nickel silicide was determined (Figure 2d). The Ni-Si alloy formed at a 300 °C annealing temperature is mainly in the Ni_2_Si phase because it is a thermodynamically favorable phase in the nickel-rich region [25]. The peaks at 26.99°, 36.49°, 47.80°, and 69.02° correspond to the (100), (002), (110), and (202) planes of Ni_2_Si, respectively, while the peaks at 23.1° and 45.84° belong to the (101) and (112) planes of NiSi. Because it is not conducive to forming silicon-rich silicide phases in a nickel-rich environment, there is an increased thermodynamic force for Si to diffuse into Ni and form NiSi phases with an increasing annealing temperature. The existence of NiSi during annealing at 400 °C is attributed to the preferential diffusion of silicon through the silicide interface and the interaction with Ni_2_Si, which makes it completely transformed into the NiSi phase [36]. The peaks at 23.1°, 31.67°, 37.82°, 45.84°, and 68.83° are attributed to the (101), (011), (201), (112), and (122) planes of NiSi, respectively, while the peaks at 56.31° and 75.59° correspond to the (311) and (331) planes of NiSi_2_. Si diffuses further and combines with NiSi to generate NiSi_2_ when the annealing temperature is raised some more. At the annealing temperature of 500 °C, the peaks at 28.6°, 56.3°, 69.35°, and 76.59° correspond to the (111), (311), (400), and (311) planes of NiSi_2_, respectively, while the peaks at 23.1°, 37.85°, and 45.84° are attributed to the (101), (201), and (112) planes of NiSi. The Ni is precisely deposited on the back of the n-TOPCon solar cells by electron beam evaporation. The Ni/Si alloy formed by annealing in a nitrogen atmosphere is a mixture, not a single compound.

The composition of the formed compound is further verified by Raman spectroscopy measurements, as shown in Figure 2e. At a 300 °C annealing temperature, the peaks at 100 cm^−1^ and 140 cm^−1^ belong to the Ni_2_Si phase. The peak at 216 cm^−1^ belongs to the phase of NiSi. At 400 °C, the NiSi phase is present at the major peak position in addition to the Ni_2_Si phase at 100 cm^−1^ [37]. It has an obvious peak intensity at 194 cm^−1^, 216 cm^−1^, 292 cm^−1,^ and 363 cm^−1^, and the peak at 396 cm^−1^ is attributed to the NiSi_2_ phase. When the annealing temperature is increased to 500 °C, the peak of the Ni_2_Si phase disappears, while the peaks at 288cm^−1^ and 396cm^−1^ belong to the NiSi_2_ phase, and the remaining peaks are in the NiSi phase. The absence of the Raman-active Si phonon band is found at 521 cm^−1^ at different annealing temperatures, which confirms that the Ni-Si alloy films formed using electron beam deposition of Ni on Si under different annealing conditions provided a continuous, crack-free film and completely covered the Si surface [38]. The NiSi thin film as the seed layer is crucial for the complete surface coverage of the silicon in the laser opening area because the presence of any defects will cause the contact resistance to be uneven, thereby reducing the solar cell performance. Based on the analysis of XRD and Raman spectra, we conclude that the Ni-Si alloy formed under different annealing conditions is a mixture, which is closest to NiSi at a 375 °C annealing temperature.

The hypothesis of the diffusion formation mechanism of nickel–silicon alloy formed by thermal silicification is proposed (Figure 3) based on the above microscopic characterization tests. At the temperature of 300 °C, Ni_2_Si would preferentially form due to the diffusion of Ni atoms to the n-poly layer. With the increase in the annealing time, the thickness of Ni_2_Si increases under the premise of ensuring the thickness of metal nickel. The Ni_2_Si with a low chemical potential, which acts as a thermodynamic driving force, would have reacted for Formula (1) [39]:(1)$$2\mathrm{Ni}+\mathrm{Si}\to {\mathrm{Ni}}_{2}\mathrm{Si}\;(-183.6\;\mathrm{kJ}\;{\mathrm{mol}}^{-1})$$

When the annealing temperature is 400 °C, both the remaining Ni and Ni_2_Si react with n-poly to form NiSi, and Ni_2_Si will gradually react with Si until it is completely or mostly consumed, which would decrease the overall thermodynamic energy. The chemical potential and reaction formula are shown in (2) and (3) [40]:(2)$$\mathrm{Ni}+\mathrm{Si}\to \mathrm{NiSi}\;\left(-115.4\;\mathrm{kJ}\;{\mathrm{mol}}^{-1}\right)$$
(3)$${\mathrm{Ni}}_{2}\mathrm{Si}+\mathrm{Si}\to 2\mathrm{NiSi}\;\left(-47\;\mathrm{kJ}\;{\mathrm{mol}}^{-1}\right)$$

When the annealing temperature is 500 °C, the Si atoms of the n-poly layer will diffuse into NiSi to form NiSi_2_. With the increase in annealing time, NiSi will be gradually consumed. The formation rate of NiSi_2_ is much slower than other silicification processes, following the chemical formula in (4) [41]:(4)$$\mathrm{NiSi}+\mathrm{Si}\to \mathrm{Ni}{\mathrm{Si}}_{2}\left(-17.4\;\mathrm{kJ}\;{\mathrm{mol}}^{-1}\right)$$

From low temperature to high temperature, the transformation order of nickel silicide is Ni_2_Si, NiSi, and NiSi_2_, which is related to the participation of Si atoms with high heat energy. The growth of thin films with only the NiSi phase is very important because the mixed film of NiSi and high resistivity the Ni_2_Si phase may lead to higher resistance-induced loss of the cells.

The thermal silicide samples at different annealing temperatures are evaluated using the TLM test of contact resistance, which is compared with the Ni/Cu/Ag contact directly electroplated without a seed layer. When using the TLM, the total resistance R_T_ between finger and finger is measured and plotted as a function of contact spacing L (shown in Figure 4a). As contact spacing L increases, the effect of the sheet resistance on the total resistance measurement increases, thus creating the slope of the TLM plot *ρ_c_* [42].

Figure 4b is the contact resistance of the samples under different annealing processes. With the increase in annealing temperature, the contact resistance of the fingers on the back of the seed layer electroplated n-TOPCon is lower than that of the reference sample. In the contact resistance test on the back of the electroplated n-TOPCon solar cell, although the overall value difference is relatively insignificant, it can still be clearly concluded that the nickel–silicon alloy formed by the annealing thermal silicification process has played a positive role in effectively reducing the contact resistance.

Using a high-resolution optical profiler, the surface roughness of several annealed samples was examined to determine how it affected the adherence of the Busbars and Fingers. Figure 4c shows the surface roughness of the seed layer under different annealing processes. From the diagram, it is shown that the surface roughness value of the sample after annealing is higher than that after laser opening the dielectric layer, which indicates that the formed nickel–silicon alloy can increase the surface roughness. The roughness of the sample annealed at 375 °C is better than that of other annealing conditions because NiSi accounts for the largest proportion of the material formed at this temperature. To evaluate the possible role of the formed Ni-Si alloy in promoting the adhesion of the gate wire, the peel force test is conducted at each temperature condition, including directly electroplated Ni/Cu/Ag contacts without a seed layer (Figure 4d). At the annealing temperature from 300 °C to 425 °C, the average adhesion values increase first and then decrease. The highest adhesion of 2.7 N mm^−1^ is recorded at an annealing temperature of 375 °C. In addition, the adhesion of the samples with different annealing processes shows more than 1 N mm^−1^ [43]. The average adhesion of the sample without the seed layer is greater than 0.8 N mm^−1^, but there are still values that do not meet the bonding force standard. The electroplating contact has a multi-layer structure. To have a beneficial gate–wire bonding force, it is necessary to improve the adhesion of the Ni-Si interface. The results show that the bonding force of the seed layer samples is better than that of the direct electroplating Ni/Cu/Ag under different annealing processes.

When UV-ps acts on the SiNx layer on the back of the seed layer n-TOPCon precursor, the Si surface is destroyed, which is manifested as melting, a heat-affected zone, microcracks, and point defects. These destructions lead to enhanced diffusion, resulting in Ni cluster defects [44,45]. The electrical properties of the seed layer n-TOPCon solar cells under different annealing processes are tested with a Sinton FCT650, and the results are shown in Table 1. For reliable data, 10 cells are measured under different annealing conditions in each group. With the increase in the annealing temperature, The Fill Factor (FF) increases first and then decreases. The average FF of the cells annealed at 375 °C is the highest, which is 82.02%, which is 1.24% higher than that of the cell annealed at 425 °C. During the annealing process, the efficiency shows a similar trend, that is, at the annealing temperature of 425 °C, the efficiency reaches the lowest level. In view of the obvious loss of FF observed at 425 °C annealing conditions, we decided not to manufacture batteries with higher annealing temperatures (Table 2).

From the results of Figure 5a, it is shown that with the increase in annealing temperature, Rs decreases first and then increases, but it is better than that of Ni/Cu/Ag directly electroplated without seed layer, which is consistent with the results of the contact resistance test. Under the condition of an annealing temperature lower than 400 °C, the pFF does not change much, so the reduction from pFF to the measured FF equals the FF loss due to Rs. Under the premise of other unchanged parameters, the relationship between Rs and FF is the opposite, and the decrease in Rs will lead to an increase in FF, so the change in FF is mainly affected by Rs (Figure 5b). It can be seen from Figure 5c that the pFF has no downward trend at 375 °C, while the pFF at 400 °C decreases by 0.6%. When the annealing temperature reaches 425 °C, the pFF reaches 81.69%, a decrease of 1%. Raval et al. [46]. showed that the pFF was not severely degraded until it reached the 400 °C annealing temperature. The composite current density (*J*0_2_) and the space charge area shunt resistance have an impact on the pFF, making it a useful metric for assessing the junction’s quality. If the metal atoms from the contacts form a trap site in the space charge region, the amount of shunt path and *J*0_2_ will be increased and finally decrease the pFF. Since the edges are protected during plating, they are not affected by any ohmic shunt. Therefore, the shunting does not adversely affect the pFF, and the loss can be safely attributed to the recombination (*J*0_2_) loss. The higher standard deviation of the average pFF value at 400 °C annealing is reflected in the recombination-based FF loss. The larger standard deviation of the average pFF value upon annealing at 400 °C represents the FF loss resulting from recombination.

## 4. Conclusions

In this study, a metal nickel layer is deposited on the backside of solar cells using electron beam evaporation, and excess nickel is removed by H_2_SO_4_:H_2_O_2_ etchant under annealing conditions of 300–425 °C to form a seed layer. XRD and Raman spectroscopy results indicate that optimizing the annealing temperature plays a crucial role in controlling the phases of the seed layer. TLM measurements showed that the specific contact resistivity of the Si/Ni stack followed the phase transition sequence with the changing annealing temperature. By annealing at 375 °C, the contact with NiSi as the primary phase was found with the lowest contact resistance. Peel strength tests along the Busbars indicated that the Si-NiSi-Ni-Cu gate line contact, prepared through annealing within the temperature range of 375 °C, exhibited superior adhesion with a peak value of 2.7 N mm^−1^. The IV measurement data indicated that the cells prepared at an annealing temperature of 375 °C exhibited the best performance, characterized by the highest FF of 82.02% and efficiency E_ff_ of 23.67%. After the stability test, the efficiency of the solar cell still remains above 90%, indicating that the prepared crystalline silicon solar cell exhibits certain stability under specific test conditions.

## Figures and Tables

**Figure 1 materials-17-02610-f001:**
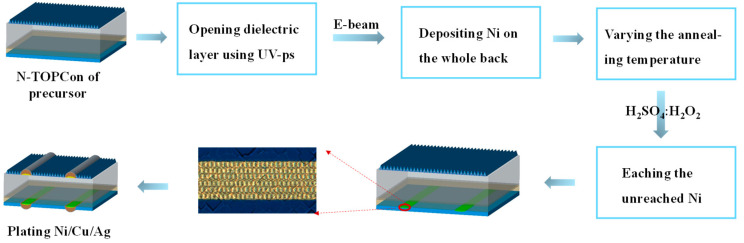
Flow chart of seed layer electroplating n-TOPCon solar cell.

**Figure 2 materials-17-02610-f002:**
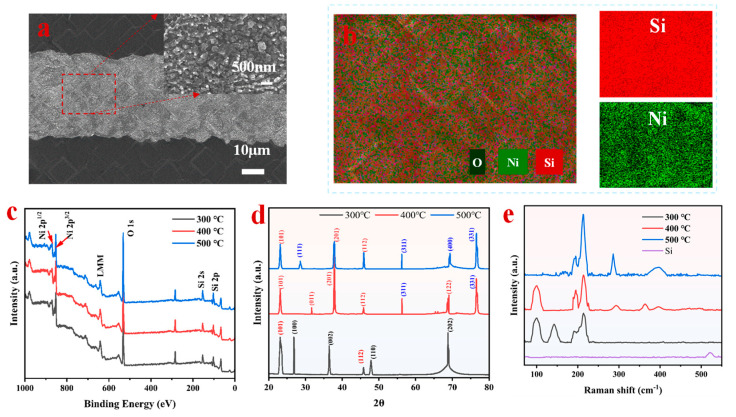
(**a**) SEM image and (**b**) EDS elements of seed layer at annealing temperature of 375 °C; the samples annealed at different temperatures of (**c**) XPS; (**d**) XRD; and (**e**) Raman.

**Figure 3 materials-17-02610-f003:**
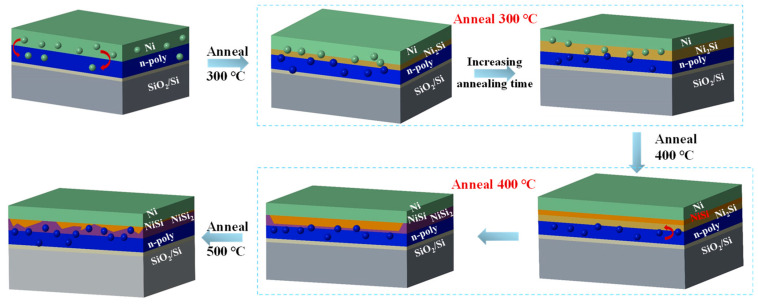
The structure diagram of Ni-Si alloy was obtained by different annealing processes. The green ball represents the Ni atom, and the blue ball represents the Si atom.

**Figure 4 materials-17-02610-f004:**
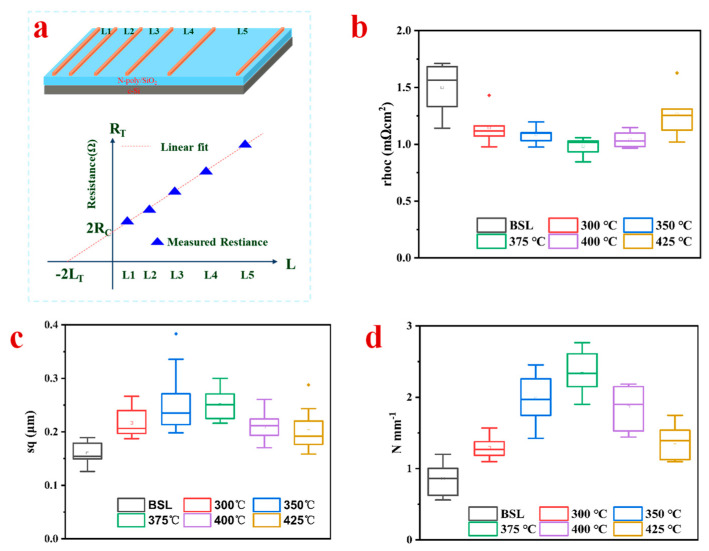
(**a**) Contact resistance model; under different annealing conditions; the samples under different annealing temperatures (**b**) contact resistance values (**c**) surface roughness of sample; (**d**) the adhesion of busbars.

**Figure 5 materials-17-02610-f005:**
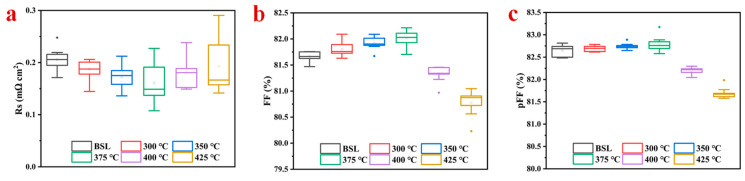
Electrical properties under different annealing conditions (**a**) Rs; (**b**) FF; and (**c**) pFF.

**Table 1 materials-17-02610-t001:** The electrical properties of the cells under different annealing processes.

Sample	Jsc (mA cm^−2^)	Voc (mV)	FF (%)	E_ff_ (%)	J0_2_ (A cm^−2^)
BSL	40.77 ± 0.14	706.53 ± 1.7	81.67 ± 0.21	23.52 ± 0.07	(6.17 ± 0.03) × 10^−9^
300 °C	40.78 ± 0.22	706.9 ± 0.7	81.8 ± 0.29	23.58 ± 0.11	(6.26 ± 0.02) × 10^−9^
350 °C	40.78 ± 0.32	706.95 ± 1.3	81.92 ± 0.27	23.62 ± 0.15	(6.08 ± 0.025) × 10^−9^
375 °C	40.82 ± 0.13	706.98 ± 1.5	82.02 ± 0.18	23.67 ± 0.12	(5.27 ± 0.015) × 10^−9^
400 °C	40.42 ± 0.35	705.67 ± 1.4	81.34 ± 0.13	23.43 ± 0.17	(10.71 ± 0.02) × 10^−9^
425 °C	39.95 ± 0.46	702.67 ± 1.8	80.78 ± 0.13	23.27 ± 0.07	(13.74 ± 0.03) × 10^−9^

**Table 2 materials-17-02610-t002:** Comparison of electrical properties before and after stability test.

Sample		Jsc (mA cm^−2^)	Voc (mV)	FF (%)	E_ff_ (%)	Rs (ohm cm^2^)
BSL	Before	40.77	706.53	81.67	23.52	0.20586
	After (4 h)	40.72	704.85	79.84	22.91	0.47449
375 °C annealing temperature	Before	40.81	706.87	81.96	23.65	0.18636
	After (4 h)	40.80	705.78	80.21	23.09	0.44138
	After (8 h)	40.44	702.12	78.56	22.31	0.81981

## Data Availability

Data are contained within the article.

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
