# Peer review of "Enhancing Adhesion and Reducing Ohmic Contact through Nickel–Silicon Alloy Seed Layer in Electroplating Ni/Cu/Ag"

_materials, 2024, doi:10.3390/ma17112610_

Round 1
Reviewer 1 Report
Comments and Suggestions for Authors
The research addresses a critical challenge in the field of solar cell production, focusing on the metallization process, which is crucial for the efficient conversion of sunlight into electricity. Traditional methods, such as screen printing with silver paste, have been effective but are associated with high costs due to the expense of silver. Seeking alternatives to reduce costs, researchers have explored various metallization techniques, with electroplating emerging as a promising approach due to its cost-effectiveness, particularly with the use of copper electrodes instead of silver.
However, electroplating comes with its own challenges, particularly concerning adhesion and ohmic contact, which are essential for the performance and durability of solar cells. In the context of Ni/Cu/Ag contacts formed by electroplating, the adhesion issue of backside grid lines in n-TOPCon solar cells poses a significant obstacle. To address this challenge, the researchers propose a novel approach: the incorporation of a Ni/Si alloy seed layer between the Ni and Si layers.
This novel approach introduces a unique solution to improve adhesion and reduce ohmic contact in the metallization process. By depositing a Ni/Si alloy seed layer and subsequently annealing it at specific temperatures, the researchers demonstrate enhanced adhesion strength and reduced contact resistance compared to traditional direct plating methods. The use of electron beam evaporation and subsequent etching to remove excess nickel during annealing further refines the seed layer, contributing to its effectiveness.
The study employs various analytical techniques, such as X-ray diffraction, Raman spectroscopy, and X-ray photoelectron spectroscopy, to characterize the phases and formation mechanisms of the Ni/Si alloy layer. Additionally, performance evaluations using Suns-Voc measurements and TLM (Transmission Line Model) measurements provide insights into the effectiveness of the proposed approach in improving cell performance.
The references are numerous and relevant.
I only spotted two spelling errors on Figure 4a, but these certainly have to be corrected before accepting the paper. The word “Resistance” is misspelled in two different ways even on the same graph. The vertical axis label is “Resitiance”, and the legend text is “Restiance”.
Since the authors must re-touch the submission, I would recommend another thing, which is only an esthetical issue. In Lines 144-146, 152-154, 156, and 158 there is a too large space before the angular degree signs. In contrast the spaces before the temperature degree Celsius signs are a bit smaller, they look better. So, I recommend reducing the space sizes at the angular degree signs (if I might miss some of them, throughout the whole text).
Overall, the research not only addresses a specific gap in the field by proposing a novel solution to the adhesion and ohmic contact challenges in electroplated Ni/Cu/Ag contacts for solar cells but also contributes to the broader goal of reducing the cost and improving the efficiency of solar cell production, thereby advancing the transition towards sustainable energy technologies.
Author Response
Dear Editor,
Thanks for your letter and for the reviewers’ comments concerning our manuscript (Manuscript ID: materials-3002701). Those comments are all valuable and very helpful for revising and improving our paper. We studied the comments carefully and revised the manuscript in accordance with the reviewers’ comments and suggestions, and carefully proof-read the manuscript to minimize grammatical and word errors. Revised portion are marked as yellow for easy checking/editing purpose.
A document answering every question from the reviewers was also summarized and enclosed as follows.
We are looking forward to your final decision on our manuscript soon, thanks!
Best regards,
Zhao Wang
School of Materials Science and Engineering
Jiangsu University,
Zhenjiang 212013,
China
E-mail: [email protected]
Corresponding author
Jianning Ding
School of Materials Science and Engineering
Jiangsu University,
Zhenjiang 212013,
China
E-mail: [email protected]
REVIEWER REPORT(S):
Reviewer #1
1 I only spotted two spelling errors on Figure 4a, but these certainly have to be corrected before accepting the paper. The word “Resistance” is misspelled in two different ways even on the same graph. The vertical axis label is “Resitiance”, and the legend text is “Restiance”.
Reply: Thanks for your feedback.
The questions raised have been checked, modified and marked in the text.
|
Figure 4: (a) Contact resistance model; under different annealing conditions; the samples under different annealing temperature (b) contact resistance values (c): surface roughness of sample; (d) the adhesion of busbars. |
2 Since the authors must re-touch the submission, I would recommend another thing, which is only an esthetical issue. In Lines 144-146, 152-154, 156, and 158 there is a too large space before the angular degree signs. In contrast the spaces before the temperature degree Celsius signs are a bit smaller, they look better. So, I recommend reducing the space sizes at the angular degree signs (if I might miss some of them, throughout the whole text).
Reply: Thanks for your feedback.
The questions raised have been checked, modified and marked in the text.

Reviewer 2 Report
Comments and Suggestions for Authors
Reviewer found some corrections should be made before it can be further considered for publication. Please revise accordingly. Comments are attached in the annotated pdf file.

English needs improvement. It is suggested inside the annotated file.
Author Response
Dear Editor,
Thanks for your letter and for the reviewers’ comments concerning our manuscript (Manuscript ID: materials-3002701). Those comments are all valuable and very helpful for revising and improving our paper. We studied the comments carefully and revised the manuscript in accordance with the reviewers’ comments and suggestions, and carefully proof-read the manuscript to minimize grammatical and word errors. Revised portion are marked as yellow for easy checking/editing purpose.
A document answering every question from the reviewers was also summarized and enclosed as follows.
We are looking forward to your final decision on our manuscript soon, thanks!
Best regards,
Zhao Wang
School of Materials Science and Engineering
Jiangsu University,
Zhenjiang 212013,
China
E-mail: [email protected]
Corresponding author
Jianning Ding
School of Materials Science and Engineering
Jiangsu University,
Zhenjiang 212013,
China
E-mail: [email protected]
REVIEWER REPORT(S):
Reviewer #2
1 is it possible to give the cost analysis of screen-printing vs plating?
Reply: Thanks for your feedback,
According to the latest market price statistics, the cost of screen printing metallized silver paste for n-TOPCon crystalline silicon solar cells is about 0.054 ¥/W. In contrast, the cost of electroplating metallization is only 0.026 ¥/W, which significantly reduces the production cost. Although the initial investment cost of screen printing equipment is lower than that of electroplating equipment, with the continuous rise of silver price and the continuous progress of electroplating technology, the price of electroplating equipment is expected to gradually decline. Therefore, copper electroplating technology is regarded as an important direction to replace screen printing silver paste in the future, and has broad market application prospects.
2 please expand N-TOPCon
Reply: Thanks for your feedback,
The questions raised have been changed and marked in the paper.
3 please quantify high amount of silver paste.
Reply: Thanks for your feedback,
The silver consumption of n-TOPCon solar cells is about 110-120 mg per piece.
4 please provide numbers to support the cost analysis
Reply: Thanks for your feedback,
According to the latest market price statistics, the cost of screen printing metallized silver paste for n-TOPCon crystalline silicon solar cells is about 0.054 ¥/W. In contrast, the cost of electroplating metallization is only 0.026 ¥/W, which significantly reduces the production cost. Although the initial investment cost of screen printing equipment is lower than that of electroplating equipment, with the continuous rise of silver price and the continuous progress of electroplating technology, the price of electroplating equipment is expected to gradually decline.
5 authors may give numbers to support their claims of conductivity prices etc.
Reply: Thanks for your feedback,
The conductivity of silver is approximately 630 S/m and that of copper is 580 S/m.
6 what is meant by merely physically connected? Does it form van der waals bond only or some ionic/covalent bond formation takes place?
Reply: Thanks for your feedback,
The metal Ni was electrochemically deposited on the Si surface without any reaction.
7 how much Ra value f roughness?
Reply: Thanks for your feedback,
The questions raised have been changed and marked in the paper.
Reference 27 states that the reasons for increased pull-off force: The application of a subsequent thermal silicidation anneal after plating can further increase the contact adhesion leading to peel forces of about 4 N mm-1 and 1.2 N mm-1 in maximum and median, respectively. The introduction of a rapid thermal anneal in the temperature range of 250-400°C can improve the contact resistivity as well as the contact adhesion.
8 it is needed to elaborate more about Kirkendall void
Reply: Thanks for your feedback,
Kirkendall void means that two metals with different diffusion rates will form defects during diffusion.
9 Please remove this portion, which is more like experimental section
Reply: Thanks for your feedback,
The questions raised have been changed and marked in the paper.
10 this sentence is more like conclusion. Does not fit in the introduction section
Reply: Thanks for your feedback,
The questions raised have been changed and marked in the paper.
11 Authors need to give details of the methods used in the study for others to reproduce the experiments, like LPCVD, PECVD, e-beam parameters. Please revise with fabrication step details or refer other work with similar processing details. Otherwise, it is incomplete information.
Reply: Thanks for your feedback.
For the preparation of n-poly layer by LPCVD (low pressure chemical vapor deposition), we adopted the conditions of deposition temperature of 600 °C and deposition time of 30 min, and finally obtained a deposition thickness of about 120 nm.
When preparing silicon nitride (SiNx) layer by plasma enhanced chemical vapor deposition (PECVD), we set the deposition temperature of 530 °C and the deposition time of 500 s. Through this process, we obtained a deposition thickness of about 100 nm.
The electron beam evaporation technique was used to prepare the nickel layer. The plating rate was set to 0.5 A s-1. After the evaporation process, we obtained a nickel layer deposition thickness of about 120 nm.
Since the process of preparing solar cells is only to prepare materials, it is not the focus of research, so there is no parameter detail of the specific parameter preparation process.
12 descripbe LIP
Reply: Thanks for your feedback.
To distribute the current in reverse bias the property of the solar cell of generating charge carriers under illumination is taken advantage of. The generated electrons are driven to the n-type doped side of the solar cell where they are reduced with metal ions whereas the holes are driven towards the p-type doped side. This process is named Light Induced Plating (LIP).
13 how FBP make no requirement of light?
Reply: Thanks for your feedback.
Conventional solar cells consist of a pn-junction so that the current is driven either in forward or in reverse bias. The plating process using forward bias to metallize the p-type doped side is defined as Forward Bias Plating (FBP)
The electrons in the p-type doping side of the n-TOPCon solar cell are provided by an external power supply during electroplating, so no light source assistance is required.
14 what is eaching?
Reply: Thanks for your feedback.
The word of eaching means using a mixture of H2SO4 and H2O2 to remove unreacted Ni in the gate region and Ni in the non-gate region.
15 optical microscope cannot measure roughness.
Reply: Thanks for your feedback.
The questions raised have been changed and marked in the paper.
The surface roughness of the seed layer was determined by high resolution optical profiler (ZetaTM-3.0).
16 please expand TLM
Reply: Thanks for your feedback.
The questions raised have been changed and marked in the paper.
Transmission Line Model (TLM)
17 authors may elaborate Suns-Voc method.
Reply: Thanks for your feedback.
In the open circuit state, the forward diffusion current of the internal forward diode is equal to the photo-generated current. The photo-generated current can be directly measured by a low-resistance ammeter (equivalent to short-circuit current). This is actually the voltage and current of the solar cell as a diode. By changing different light intensities, the IV characteristic curve of the diode without considering the series resistance Rs is actually obtained. By changing different light intensities, the analysis is obtained, so it is called Suns ( different light intensities ), and different Vocs are obtained, that is, Suns-Voc. During the experimental test, we set the light intensity to 1 Sun, and the only Voc was obtained. According to the calculation of FCT650's own equipment, the IV characteristics of solar cells, such as Isc, FF, Eff, etc., are obtained.
18 what is figure S1?
Reply: Thanks for your feedback.
Figure S1 shows the surface morphology of different annealing temperatures.
|
Figure S1: SEM images of samples at different annealing temperatures. |
19 please be more scientific, SEM image contrast does not show unevenness of surface, rather AFM does show roughness of surface
Reply: Thanks for your feedback.
The questions raised have been changed and marked in the paper.
Figure S1 shows the surface morphology of different annealing temperatures, all of which show rough graininess.
20 what is laser slotting? what is meant by melting state? Please use more logical and scientific terms.
Reply: Thanks for your feedback.
The questions raised have been changed and marked in the paper.
The production of nickel-silicon alloy compounds or the ablation state brought on by laser process could be the origin of this morphology, which would increase the surface roughness.
21 where is the XRD, Raman and XPS data for sample annealed at 375 oC?
Reply: Thanks for your feedback.
Figure 2a and b are the SEM and EDS images of the seed layer annealed at 375 °C. Figure (c-e) are XRD, Raman and XPS images at three annealing temperatures (300 °C 400 °C and 500 °C).
22 how do you know NiSi phase is the lowest resistive?
Reply: Thanks for your feedback.
The resistivity of continuous thin films was measured by four-point probe measurement The resistivity of Ni2Si, NiSi and NiSi2 are 22.80 µΩ∙cm, 14.13 µΩ∙cm and 37.77 µΩ∙cm, respectively [31].
23 what logic author tried to make here?
Reply: Thanks for your feedback.
Through X-ray photoelectron spectroscopy (XPS) analysis, we have confirmed that the prepared seed layer contains Ni and Si elements. In order to further explore and specify the material structure of these elements in the seed layer, we performed X-ray diffraction (XRD) tests. The main purpose of this test is to further verify whether the seed layer is indeed composed of nickel-silicon alloy.
24 what is the JCPDS file no.with which the data was compared to understand the phases?
Reply: Thanks for your feedback.
Ni2Si: PDF#03-1069
NiSi: PDF#38-0844
NiSi2: PDF#43-0989
25 all phases are some kind of silicides, please be uniform and consistent in naming different phases
Reply: Thanks for your feedback.
The questions raised have been changed and marked in the paper.
26 please rewrite the sentence, it is not clear what authors try to say here.
Reply: Thanks for your feedback.
The questions raised have been changed and marked in the paper.
The Ni is precisely deposited on the back of n-TOPCon solar cells by electron beam evaporation. The Ni/Si alloy formed by annealing in a nitrogen atmosphere is a mixture not a single compound.
27 there are 2 peaks around 216, why?
Reply: Thanks for your feedback.
At 400 °C, the obvious peak intensity at 194 cm-1 and 216 cm-1 (2 peaks around 216 cm-1) are attributed to NiSi phase.
28 please improve English writing style.
Reply: Thanks for your feedback.
The questions raised have been changed and marked in the paper.
29 No, where is it at 521 cm-1 in each Raman graph?
Reply: Thanks for your feedback.
At different annealing temperatures, the Ni/Si alloy seed layer is uniformly and completely covered on the Si surface, and no gaps or defects are observed. Therefore, in the Raman test, the peak intensity of Si could not be detected.
30 what kind of defect?
Reply: Thanks for your feedback.
If the coverage of the Ni/Si alloy seed layer is incomplete or there is a gap, this will lead to uneven distribution of the gate line in the subsequent metallization process, which will significantly affect the distribution of the current. This uneven current distribution will directly have a negative impact on the series resistance (Rs)of crystalline silicon solar cells, and ultimately reduce the overall efficiency of solar cells. Therefore, ensuring the integrity and seamless coverage of the Ni/Si alloy seed layer is one of the keys to improving the performance of solar cells.
31 But authors do not have XRD, Raman and XPS for sample annealed at 375 oC, how do they claim then that it is the optimal processing temp.?
Reply: Thanks for your feedback.
After microscopic test characterization, we found that when the annealing temperature is 400 °C, the proportion of NiSi in the Ni/Si alloy seed layer reaches the maximum. Based on this finding, in the process of preparing crystalline silicon solar cells, we further refined the annealing temperature near 400 °C to prepare the seed layer, and tested the electrical properties of the obtained solar cells.
From the electrical performance test results, when the annealing temperature is set to 375 °C, the electrical performance of the solar cell is the best. However, with the increase of annealing temperature, the electrical properties decreased significantly when the temperature reached 425 °C. In view of the above results, we decided not to verify the annealing conditions higher than 425 °C.
In summary, by carefully optimizing the annealing temperature, we successfully prepared crystalline silicon solar cells with excellent electrical properties, especially under the annealing condition of 375 °C, its performance reached the best.
32 please write chemical formula correctly
Reply: Thanks for your feedback.
The questions raised have been changed and marked in the paper.
Figure 2: (a) SEM image and (b) EDS elements of seed layer at annealing temperature of 375 °C; the samples annealed at different temperatures of (c) XPS; (d) XRD; (e) Raman. |
33 where is 375 oC annealing temp.?
Reply: Thanks for your feedback.
Figure 2a and b are the SEM and EDS images of the seed layer annealed at 375 °C. Figure (c-e) are XRD, Raman and XPS images at three annealing temperatures (300 °C 400 °C and 500 °C).
Figure 2: (a) SEM image and (b) EDS elements of seed layer at annealing temperature of 375 °C; the samples annealed at different temperatures of (c) XPS; (d) XRD; (e) Raman. |
34 what is meant by remaining?
Reply: Thanks for your feedback.
The word of remaining means unreacted Ni.
35 did the authors successively annealed samples at higher temperatures or different samples were annealed at different temperatures? If authors anneal a sample at 400 oC and if they re-anneal a previously annealed sample at 300 oC to higher temperature of 400 oC, it would have different effect altogether.
Reply: Thanks for your feedback.
In the experimental process described in this paper, the annealing treatment is carried out in a single continuous way, that is, the secondary annealing method of first low temperature and then high temperature is not adopted. This decision is based on the consideration of actual production costs, because secondary annealing will undoubtedly increase additional manufacturing costs. In future studies, we will fully consider your suggestions and explore the effect of secondary annealing on the performance of Ni/Si alloy seed layer and solar cells. Thank you for your valuable comments, which will provide important direction for our follow-up research.
36 but is it okay to have NiSi2 phase mixed with NiSi? what would be its resistance?
Reply: Thanks for your feedback.
At the annealing temperature of 500 °C, the NiSi alloy seed layer is a mixture of NiSi and Ni2Si phases, which is also proved by XRD and Raman measurement. The resistivity of the mixture was 25.32 µΩ∙cm by four-point probe measurement.
37 authors kept on heating at higher temperatures or they used separate sample to heat at different temperatures because in the structure diagram one of the main criterion "the annealing time" is absent. Please also include TIME into the structure diagram.
Reply: Thanks for your feedback.
In the experiment described in this paper, we uniformly used an annealing time of 90 s for different annealing temperatures. At the same annealing temperature, if the annealing time is prolonged and the supply of Si and Ni is sufficient, the thickness of the formed compound layer will gradually increase. In order to keep the focus of the discussion and simplify the chart expression, we do not clearly indicate the specific annealing time in the structure diagram. This treatment aims to highlight the main effect of annealing temperature on the formation and properties of the Ni/Si alloy seed layer.
38 please also separately write names adjacent to the figure of different color particles and how are they diffusing into the other layer to form the successive phases?
Reply: Thanks for your feedback.
The questions raised have been changed and marked in the paper.
At the temperature of 300 °C, Ni2Si would preferentially formed due to the diffusion of Ni atoms to the n-poly layer. With the increase of annealing time, the thickness of Ni2Si increases under the premise of ensuring the thickness of metal nickel. the Ni2Si with a low chemical potential, which acts as a thermodynamic driving force. When the annealing temperature is 400 °C, both the remaining Ni and Ni2Si react with n-poly to form NiSi, and Ni2Si will gradually react with Si until it is completely or mostly consumed, which would decrease the overall thermodynamic energy. When the annealing temperature is 500 °C, the Si atoms of the n-poly layer will diffuse into NiSi to form NiSi2. With the increase of annealing time, NiSi will be gradually consumed. The formation rate of NiSi2 is much slower than other silicification processes.
39 at which L it was measured?
Reply: Thanks for your feedback.
In the design of solar cells, the finger distance is marked as L, which is the preset spacing between the metallized grid lines. Among them, L1 refers to the width of a single finger, and so on, L5 represents the total width of the continuous arrangement of five fingers.
40 please check English grammar
Reply: Thanks for your feedback.
The questions raised have been changed and marked in the paper.
41 reduction is insignificant
Reply: Thanks for your feedback.
The questions raised have been changed and marked in the paper.
42 reference sample of direct electroplating without seed layer
Reply: Thanks for your feedback.
BSL is a reference sample for direct plating without seed layer.
43 what is the unit of y-axis? how much is the contact resistance?
Reply: Thanks for your feedback.
The unit of the y-axis is Ohm.
Figure 4a is the contact resistance test model. The results of the specific contact resistivity are shown in Figure 4b.
44 what is the unit in x-axis? how much are L values?
Reply: Thanks for your feedback.
The unit of the x-axis is μm. L is 1.448 μm.
45 this is wrong method of roughness estimation by microscope.
Reply: Thanks for your feedback.
The questions raised have been changed and marked in the paper.
The surface roughness of the seed layer was determined by high resolution optical profiler (ZetaTM-3.0).
46 write formula correctly
Reply: Thanks for your feedback.
The questions raised have been changed and marked in the paper.
47 wrong statement, figure 4c shows annealing at 425oC produce lowest roughness
Reply: Thanks for your feedback.
The surface roughness of the sample annealed at 375 °C is significantly superior to other annealing conditions, and it is showed that the sample annealed under this condition performs better in terms of gate-line bonding force.
48 Please revise English
Reply: Thanks for your feedback.
The questions raised have been changed and marked in the paper.
49 where?
Reply: Thanks for your feedback.
The questions raised have been changed and marked in the paper.
In the test of reference sample of direct electroplating without seed layer, we found that the average adhesion was more than 0.8 N mm-1.However, the binding force of some PAD points still failed to meet this standard, which was lower than 0.8 N mm-1. There refers to these PAD points with weak binding performance.
50 authors need to elaborate how it happens
Reply: Thanks for your feedback.
In this paper, we use UV-ps laser to process the back of the solar cell, which aims to remove the dielectric layer on the back. However, it is worth noting that under the influence of laser heat, the removal of the dielectric layer will also have a certain effect on the poly layer. Although your suggestions on the mechanism of laser action are very enlightening, they are not the main research focus of this paper. The core of this paper is to explore the effect of Ni-Si alloy seed layer on the performance of crystalline silicon solar cells. For the laser action mechanism you mentioned, we will make a more detailed discussion in the follow-up study and introduce it in another small paper. Thank you for your valuable suggestions, which will provide important guidance for our follow-up work.
51 please provide image proofs
Reply: Thanks for your feedback.
Figure 3: SEM image after back laser processing |
52 expand FF
Reply: Thanks for your feedback.
The questions raised have been changed and marked in the paper.
53 please improve English
Reply: Thanks for your feedback.
The questions raised have been changed and marked in the paper.
54 not really.....
Reply: Thanks for your feedback.
According to the IV test results, the average series resistance (Rs) data is shown in Figure 5a. It can be seen from the data analysis of figure 5a that as the annealing temperature gradually increases, the Rs value first shows a decreasing trend, and then gradually increases. However, all the annealed samples exhibited better Rs performance than the samples directly electroplated without seed layer. This result is consistent with the results of the contact resistance test, which further verifies the effectiveness of annealing treatment to improve the resistance performance.
55 authors need to elaborate how it happens
Reply: Thanks for your feedback.
When the annealing temperature is lower than 400 °C, the value of pFF (i.e., the fill factor ignoring the effect of series resistance Rs) remains relatively stable, which indicates that the decrease of FF is not directly caused by the change of pFF. Therefore, it can be inferred that the decrease of FF is mainly caused by the increase of Rs.
56 Authors have abruptly used few terms, like, composite current density (composite of what?), trap site, shunt resistance, shunt path etc., without elaborating on the scientific phenomenon behind it. They must revise accordingly.
Reply: Thanks for your feedback.
The composite current density (J02) is one of the key indicators to evaluate pFF (FF ignoring the influence of series resistance Rs ). It will be accurately measured in the IV test of crystalline silicon solar cells. In addition to the series resistance Rs, the parallel resistance is also an important factor affecting the change of the FF. Prior to the start of the electroplating process, we have taken edge protection measures designed to prevent the reduction of the parallel resistance, thereby preventing the decrease of the FF. The above contents are one of the important indexes to evaluate the performance of solar cells.
57 annealing time is very important parameter as the silicification is diffusion controlled phenomenon, it is not only temperature dependent but also time dependent. Therefore, more data should be given with consideration of annealing time as parameter at different annealing temperatures (not only at 375 oC)
Reply: Thanks for your feedback.
In the experiment described in this paper, we uniformly used an annealing time of 90 s for different annealing temperatures.
In the follow-up study, we will pay close attention to this issue and explore the effect of annealing time on the performance of solar cells. By increasing the annealing time, the thickness of the Ni/Si alloy seed layer will increase, but this will inevitably consume more poly-Si materials. Poly-Si plays an important passivation role in n-TOPCon solar cells. However, when more Si is consumed, the poly layer in the gate region will become weaker, and its passivation performance will also decrease, which may lead to a decrease in the efficiency of solar cells.
Therefore, in future studies, we will re-examine and optimize the effect of annealing time, while reconfiguring its structure when preparing solar cells. The possible strategy is to thicken the poly layer in the gate line region to ensure sufficient Si source for the synthesis of nickel-silicon alloy, thus ensuring the efficiency of solar cells.
58 Please revise the manuscript as suggested with annotated comments from the reviewer. Thanks.
Reply: Thanks for your feedback.
The questions raised have been changed and marked in the paper.
